# Dynamic Monitoring of Nutrition Inputs and Fertility Evaluation during a Decade in the Main Peach-Producing Areas of Shandong Province, China

**DOI:** 10.3390/plants12081725

**Published:** 2023-04-20

**Authors:** Tingting Li, Binbin Zhang, Anqi Du, Sankui Yang, Kexin Huang, Futian Peng, Yuansong Xiao

**Affiliations:** College of Horticulture Science and Engineering, Shandong Agricultural University, Taian 271018, China

**Keywords:** peach orchard, soil fertility, dynamic change, nutrition inputs

## Abstract

The main peach-producing area in Shandong is an important peach fruit-producing area in China. Understanding the nutritional properties of the soil in peach orchards helps us to understand the evolution of soil properties and adjust management methods in a timely manner. This study focuses on 52 peach orchards in the main peach-producing area in Shandong as the research object. The spatiotemporal changes in soil traits and their influential factors were studied in depth, and the changes in soil fertility were effectively evaluated. The results showed that the input of nitrogen, phosphorus and potassium from organic fertilizer in 2021 was significantly higher than that in 2011, while the input of fertilizer in 2011 was significantly higher than that in 2021. Compared with traditional parks, both organic fertilizer inputs and chemical fertilizer inputs in demonstration parks showed a significant downwards trend. There was no significant change in pH values between 2011 and 2021. In 2021, the soil organic matter (SOM) contents of the 0–20 cm and 20–40 cm layers were 24.17 g·kg^−1^ and 23.38 g·kg^−1^, respectively, an increase of 29.3% and 78.47% over the values measured in 2011. Compared with 2011, the content of soil alkaloid nitrogen (AN) decreased significantly in 2021, and the contents of available phosphorus (AP) and available potassium (AK) in the soil increased significantly. According to the calculation results of the comprehensive fertility index (IFI) value, we found that in 2021, compared with 2011, the quality of soil fertility improved, most of which was at the medium and high levels. The research results show that the fertilizer-saving and synergistic approach in peach orchards in China significantly improved the soil nutrition. In the future, research on suitable comprehensive technologies should be strengthened in the management of peach orchards.

## 1. Introduction

Soil is the material support on which humans depend for survival and an important way for plants to obtain nutrients and moisture, and it plays an indispensable role in the natural ecosystem [1,2]. Soil nutrients are an important regulator in the process of plant growth and an important indicator for evaluating soil fertility [3,4]. The abundance and shortage of soil nutrients represented by soil organic matter (SOM), soil alkaloid nitrogen (AN), available phosphorus (AP) and available potassium (AK) have the most direct and far-reaching impacts on agricultural production and food security [5,6].

Due to the combined effect of many natural factors, such as the terrain, soil parent material, climate and human activities, soil nutrients have certain spatial variability [7,8,9]. Therefore, an accurate understanding of the spatial variation characteristics of soil nutrients and their influential factors is of great significance to promote scientific fertilization, improve the level of agricultural production and operation, and increase soil fertility [10,11].

In recent years, there have been many studies on the spatial variation characteristics of soil nutrients locally and abroad, some involving large-scale studies such as countries and watersheds and some involving small-scale studies such as orchards and fields [12]. The nutrient indicators selected in the research were focused on SOM and AN, AP, AK and other key elements that determine crop growth [13,14,15]. Research methods mostly use classical statistical analysis, geostatistical analysis, correlation analysis and other methods [16]. Previous studies have shown that the environmental factors that restrict the spatial variation in soil nutrients at different research scales are different. On a large scale, soil nutrients are mainly affected by topography and climate. For example, climate change will have an impact on the soil structure and soil fertility of orchards, which will further affect the annual reproductive cycle of fruit trees; the nutrient rest period of fruit trees is shortened, resulting in the early flowering of fruit trees. On a small and medium scale, human activities are the leading factor affecting the physical and chemical properties of the soil [17,18]. In the process of agricultural planting, people cultivate the soil. Applying soil fertilizer can protect the soil and ecological environment, meet the needs of various nutrient elements in the process of crop growth, and ensure the healthy growth of crops.

However, as a large agricultural country, China has been applying many fertilizers and nitrogen fertilizers for a long time, which has led to a series of problems, such as serious soil nutrient imbalances, soil compaction, acidification, and soil fertility decline, which not only seriously affect the yield and quality of crops but also cause environmental pollution [19,20,21]. In recent years, with the continuous development of modern agriculture, ‘zero fertilizer growth’ has become the main goal of modern agricultural fertilization [22]. Fertilization methods have also changed. For example, the slow-control technology of fertilizer can control nutrient release, stabilize the nutrient supply, and better solve the problems caused by the application of quick-acting fertilizers [23]. However, due to its high price, it is difficult to popularize it on a large scale in the production of fruit trees. Bag-controlled slow-release fertilizers changed the previous design ideas. According to the characteristics of the larger fruit tree body, controlled-release bags were used to control nutrient release to achieve consistent fertilizer supply and nutrient requirements, with a low cost and simple process more conducive to large-scale popularization and application [24,25]. These changes in fertilization methods not only reduce the amount of chemical fertilizer input but also improve the soil fertility and increase the yield and quality of crops.

China is the world’s largest producer of peaches, with a history of cultivation of 3000 years. At present, its production and area account for more than 50% of the global values, and the planting area of peach trees covers almost all provinces [26,27]. In China, the excessive use of chemical fertilizers has led to serious environmental degradation, and the amount of chemical fertilizers used in orchards is increasing each year. There is a serious problem of excessive nutrient input in the production and management of peach orchards [28]. Therefore, a better understanding of the spatiotemporal changes in soil properties in peach-producing areas in our country will help guide the sustainable development of our peach industry. We believe that under different fertilization conditions, soil fertility will change with the amount of fertilizer. In this study, we monitored the spatiotemporal variation in soil physical and chemical properties and soil fertility under different fertilization conditions in the same area over one decade. The classical statistical analysis method was used to describe and analyze the degree of variation in the soil chemical properties to facilitate the generation of reliable and accurate mean estimates under a specific sample size and reasonable sample layout to provide data support for the effective management of regional soil [29,30,31]. This would more intuitively reflect the characteristics of soil fertility changes in peach orchards, comparing ten years ago to ten years later, and enable us to compare the results with the demonstration peach orchard to highlight the effects of weight loss and fertilizer control technology and provide a strong basis for subsequent peach-orchard fertilization management. The fuzzy comprehensive evaluation method was used to quantitatively evaluate soil fertility, which can more objectively reflect the comprehensive fertility status of the soil [32,33].

The purpose of this research was: (i) to study the changes in soil properties over one decade, (ii) explore the causes of the soil property changes and (iii) evaluate the changes in soil fertility in peach-producing areas in China.

## 2. Results

### 2.1. Analysis of the Physical and Chemical Properties of Soil at Different Times and in Different Locations

The pH values of Tai’an (TA) in 2011 and 2021 were 6.51 and 6.55, and those of the Meng’yin (MY) production area were 6.3 and 6.3, respectively (Figure 1). The average pH value of the main peach-producing areas in Shandong was 6.42 over the decade, and there was no significant change in the overall value.

The effective nutrient content of orchard soil in the main peach-producing area of Shandong in 2021 is presented in Table 1. The SOM content in the 0–20 cm soil layer was 24.17 g·kg^−1^, which is an increase of 29.3% compared with that in 2011. The SOM contents of the TA production area and MY production area were 25.61 g·kg^−1^ and 22.2 g·kg^−1^, respectively. The SOM content in the 20–40 cm soil layer was 23.38 g·kg^−1^, which is an increase of 78.47% compared with that in 2011. The SOM contents of the TA production area and MY production area were 23.8 g·kg^−1^ and 22.82 g·kg^−1^, respectively (Figure 2A and Figure 3A). The AN content in the 0–20 cm soil layer was 83.16 g·kg^−1^, which was 3.12% lower than that in 2011. The soil AN contents of the TA production area and MY production area were 82.75 mg·kg^−1^ and 83.73 mg·kg^−1^, respectively. The AN content in the 20–40 cm soil layer was 65.56 mg·kg^−1^, which is a decrease of 31.89% compared with that in 2011. The AN contents of the TA production area and MY production area were 55.2 mg·kg^−1^ and 79.68 mg·kg^−1^, respectively (Figure 2B and Figure 3B). The AP content in the 0–20 cm soil layer was 73.68 mg·kg^−1^, which was 8.57% higher than that in 2011. The AP of the TA production area and MY production area was 90.54 mg·kg^−1^ and 50.68 mg·kg^−1^, respectively. The AP content in the 20–40 cm soil layer was 43.11 mg·kg^−1^, i.e., 16.6% lower than that in 2011. The AP contents in the TA production area and MY production area were 51.16 mg·kg^−1^ and 32.14 mg·kg^−1^, respectively (Figure 2C and Figure 3C). The AK content in the 0–20 cm soil layer was 256.54 mg·kg^−1^, which was 182.22% higher than that in 2011. The AK contents in the TA production area and MY production area were 254.52 mg·kg^−1^ and 259.29 mg·kg^−1^, respectively. The AK content in the 20–40 cm soil layer was 233.02 mg·kg^−1^, which was 311.47% higher than that in 2011. The AK contents of the TA production area and MY production area were 213.45 mg·kg^−1^ and 236.06 mg·kg^−1^, respectively (Figure 2D and Figure 3D).

### 2.2. Analysis of the Physical and Chemical Properties of Soil in the Demonstration and Traditional Peach Orchards

In the demonstration peach orchard, measures were implemented to reduce fertilizer and fertilizer control to increase the soil organic matter content and improve soil fertility. In traditional peach orchards, conventional fertilization methods are used to improve soil fertility. Based on the comparison of the soil nutrient content of more than 100 traditional peach orchards and 6 demonstration peach orchards surveyed (Figure 4), SOM, AP and AK in the traditional park did not differ significantly when the amount of fertilizer differed. The content of AN in the traditional peach orchards was not significantly different from that in demonstration peach orchards 1 and 2, but was significantly different from that in demonstration peach orchard 3.

### 2.3. Correlation Analysis of Soil Indicators

The correlation analysis of the soil indicators is shown in Figure 5. In 2011, the SOM in the 0–20 cm layer was negatively correlated with AN and positively correlated with AP and AK. AN was positively correlated with AP and AK and extremely positively correlated with AP. There was a very positive correlation between AP and AK. In 2011, the SOM in the 20–40 cm layer was positively correlated with AN, AP and AK; AN was negatively correlated with AP and AK; and there was a very positive correlation between AP and AK. In 2021, the SOM in the 0–20 cm layer was positively correlated with AN and AK and negatively correlated with AP; AN was negatively correlated with AP and AK; and AP was positively correlated with AK. In 2021, the SOM in the 20–40 cm layer was positively correlated with AN and negatively correlated with AP and AK (very negative correlation); AN was negatively correlated with AP and very positively correlated with AK; and AP was positively correlated with AK.

### 2.4. Classification of the Soil Effective Nutrient Content

According to China’s second soil census standard, the soil nutrient status is divided into six levels: enriched sample proportion (1), appropriate sample proportion (2), medium sample proportion (3), low sample ratio (4), lower sample ratio (5), and extremely low sample proportion (6). The two soil layers in the main peach-producing areas of Shandong Province were graded over the decade (Figure 6). In the 0–20 cm soil layer, in 2021, compared with 2011, the SOM content at the medium and higher levels accounted for 86.54%, an increase of 59.62%; the AN content accounted for 34.62%, a decrease of 3.84%; AP accounted for 96.15%, a decrease of 1.92%; and AK accounted for 96.15%, an increase of 63.46%. In 2021, compared with 2011, the SOM content at below the medium level accounted for 13.46%, a decrease of 59.62%; AN accounted for 65.38%, an increase of 3.84%; AP accounted for 3.85%, an increase of 1.92%; and AK accounted for 3.85%, a decrease of 63.46%. In the 20–40 cm soil layer, compared with 2011, the SOM content at the medium and higher levels accounted for 65.38% in 2021, an increase of 59.61%; AN accounted for 17.3%, a decrease of 28.85%; AP accounted for 88.47%, an increase of 5.78%; and AK accounted for 100%, an increase of 88.46%. In 2021, compared with 2011, the SOM content below the medium level accounted for 34.62%, a decrease of 59.61%; AN accounted for 82.69%, an increase of 28.84%; AP accounted for 11.54%, a decrease of 5.77%; and AK accounted for 0%, a decrease of 88.46%.

### 2.5. Analysis of the Difference in the Fertilizer Volume

The total inputs of fertilizers, organic fertilizers, and nutrients in the demonstration parks of the two production areas over the decade are shown in Table 2. The input of nitrogen, phosphorus and potassium in organic fertilizers in 2021 was significantly higher than that of nitrogen, phosphorus and potassium in 2011, while the input of fertilizer in 2011 was significantly higher than that in 2021. Compared with the traditional parks, both organic and chemical fertilizer inputs in the demonstration parks showed a significant downwards trend.

### 2.6. Comprehensive Evaluation of Soil Fertility

As shown in Table 3, the membership values of SOM, AN, and AK in the 0~20 cm soil layer in 2011 were all less than 0.5, which shows that SOM, AN, and AK were the main factors limiting soil fertility. The membership values of SOM, AN and AK in the 20~40 cm soil layer were also less than 0.5, which restricted the soil fertility, among which SOM and AK were the most obvious. In 2021, only the subordination value of AN in the 0~20 cm soil layer was slightly less than 0.5, which indicates that AN had a limiting effect on soil fertility; in the 20~40 cm soil layer, the AN was also less than 0.5, which shows that the two major soil layers in 2021 were most obviously restricted by AN, while other indicators did not play a restrictive role in soil fertility.

Based on the calculation results of the IFI value, the results of the fertility and quality levels of the orchards in all surveyed areas over the decade were analyzed, as shown in Table 4. The analysis results in Table 4 show that in the soil layer of 0–20 cm in the main peach-producing area of Shandong Province in 2021, compared with 2011, the orchards with soil fertility quality at levels I and II accounted for 38.47%, an increase of 30.78%, and the soil fertility quality was at a higher level; orchards at level III accounted for 51.92%, an increase of 13.46%; orchards at level IV accounted for 9.62%, a decrease of 30.76%, and the soil fertility quality was poor; and orchards at level V accounted for 0%, a decrease of 13.46%. This shows that the soil fertility quality in 2011 was mostly at a medium-low level, while in 2021, the soil fertility quality was mostly at a medium-high level. In the 20–40 cm soil layer, orchards with soil fertility quality at levels I and II in 2021 accounted for 23.07%, an increase of 19.22% compared to 2011, and the soil fertility quality was at a higher level; orchards at level III accounted for 65.38%, an increase of 55.76%, and the soil fertility quality was medium; orchards at level IV accounted for 9.62%, a decrease of 57.7%; and orchards at level V accounted for 1.92%, a decrease of 17.31%, and the quality was poor. This shows that most of the soil fertility quality in 2011 was at a poor level, compared to a medium to high level in 2021.

## 3. Discussion

In this study, we monitored the spatiotemporal variation in soil physical and chemical properties and soil fertility under different fertilization conditions in the same area over one decade. Soil pH is a key indicator for evaluating soil quality and plays an important role in improving soil fertility and soil physical and chemical properties and in promoting plant growth [34,35]. The monitoring results showed that the soil pH value did not change significantly over the decade. In 2021, compared with 2011, it increased by only 0.05 units. Previous studies have shown that regardless of natural conditions or agricultural measures, the process of soil pH change is very slow [36]. With the change in fertilization methods, the amount of fertilizer input has been greatly reduced. This may be the reason why the pH value of the main peach-producing area in Shandong remained unchanged or increased slightly.

Through descriptive analysis, we learned that in 2011 and 2021, the soil nutrient content of the 0–20 cm soil in the studied area was higher than that of the 20–40 cm soil [37]. The reason may be that the surface of the soil is 0–20 cm, which is the interface between the soil and the atmosphere. The soil formation effect is the strongest and the time is the longest. It has strong bioaccumulation, contains more humus, and has high fertility. For cultivated soil, it is also the location of fertilization that will naturally cause it to be the most nutrient-rich.

The contents of SOM, AP and AK increased over the decade. This is mainly related to the progression of orchard cultivation in the past decade. Traditional cultivation modes include grass cutting, grazing, fertilization and farming [38,39]. However, the long-term implementation of these traditional models will cause environmental problems such as soil erosion, a weakened carbon sequestration capacity, and nitrate pollution [40]. The modern cultivation model adopts management methods such as orchard grass production, fertilizer reduction, fertilizer control and nano-fertilizers, which can not only increase the diversity of soil microbial communities but also improve the soil nutrient content and soil fertility [41,42,43,44,45].

The content of AN decreased over the decade, which is consistent with a previous study [46]. Studies have shown that approximately 50% of global N_2_O emissions are related to the application of agricultural soil nitrogen fertilizer [47]. As the largest consumer of nitrogen fertilizer, China accounts for approximately 36% of the world’s fertilizer use. Although the use of nitrogen fertilizer in peach orchards has always been high, the utilization rate of peach trees is very low [48,49]. In recent years, slow-release fertilizers have gradually replaced the traditional method of applying nitrogen fertilizers in large quantities. On the basis of achieving the same benefits, they not only reduce the input of nitrogen fertilizers but also effectively supply the soil with various required nutrients and improve crop yield and quality [50].

SOM and nitrogen were negatively correlated in 2011, but SOM and nitrogen were positively correlated in 2021, which shows that proper nitrogen control within a certain range can increase the content of SOM [51].

Compared with pH, AN and SOM, the content of AK had high variability (182~311%). Potassium is the most important nutrient in the soil, but the content of potassium in the soil in different regions will change with changes in the soil physical and chemical properties [52]. Organic fertilizers can significantly increase the potassium content in the soil. In recent years, people have reduced their investment in fertilizers and have begun to apply a large number of organic fertilizers and biological fertilizers. Moreover, many organic fertilizers are derived from manure produced by farmers’ own cattle, sheep, chickens, pigs, etc. This may be the reason for the high potassium content in the soil [53,54,55].

As seen from our research, the input of organic nitrogen, phosphorus and potassium fertilizer in 2021 was significantly higher than that in 2011, while the input of fertilizer in 2011 was significantly higher than that in 2021 (Table 2). Compared with traditional peach orchards, the input of organic fertilizer and fertilizer in the demonstration peach orchard showed a significant downwards trend. However, there was no significant difference between the nutrient content of the demonstration peach orchard and the nutrient content of the traditional peach orchard. We speculate that there were some changes in the microbial content in the soil due to the decrease in fertilization. Some microbial flora, such as those involved in nitrogen fixation and phosphorus and potassium decomposition in the soil, are active in the soil [56]. These are still the focus of our team’s next research study.

By reducing the heavy use of fertilizers, establishing a sufficient nutrient balance between fertilizer input and crop nutritional needs is a necessary measure to effectively improve soil fertility, and it is also a problem that urgently needs to be solved in agriculture [57]. This research focuses on 52 peach orchards in the main peach-producing area in Shandong. The spatiotemporal changes of soil traits and their influencing factors were studied in depth, and the changes in soil fertility were effectively evaluated. It was found that under the management conditions of fertilizer control and weight loss, the soil nutrient content of the demonstration park was not significantly different from that of the traditional park. It is concluded that water-saving and efficiency-enhancing measures have significantly improved soil nutrients. This provides theoretical support for future research on peach-orchard management technology. However, there have been some changes in whether reducing fertilization affects the content of microorganisms in the soil. Whether it promotes some microbial flora, such as those involved in soil nitrogen fixation and phosphorus and potassium decomposition, to become active in the soil, thereby improving soil fertility, we still have not reached a conclusion. This is still the focus of our team’s next research study. Although the soil fertility of the main peach-producing area in Shandong increased over the examined decade, overall, there was still a nutrient imbalance in the soil. This may be because although farmers reduced the amount of fertilizer in their daily management, the proportion of fertilizer was still unbalanced, resulting in the content of certain elements in the soil being too high or too low. According to our daily visits, we learned that farmers often pay more attention to the application of a large number of elements and often easily ignore the application of trace elements. Therefore, the adoption of a balanced fertilization strategy according to local conditions is recommended to promote the balance between a large number of soil elements and medium and trace elements and to guarantee the long-term development of the soil.

## 4. Materials and Methods

### 4.1. Location Description

This study was conducted in the main peach-producing area of Shandong Province, China (114°19′–122°43′ E, 34°22′–38°23′ N), and representative peach orchards under conventional local management conditions were selected. The demonstration peach orchard mainly adopts measures to reduce fertilizer and fertilizer control to increase soil organic matter content and improve soil fertility. Traditional peach orchards use conventional fertilization methods to improve soil fertility. This area is characterized by a temperate monsoon climate, with an average annual precipitation of 700 mm, an average annual temperature of 14 °C, and mountainous hills. The test season in 2011 was autumn. During the test period, the average temperature was 15 °C, and the precipitation was 737.1 mm. The test season in 2021 was also autumn; the average temperature during the test period was 14.5 °C, and the average precipitation was 748.2 mm. The main peach-producing area in Shandong is an important peach-producing area in China. With the continuous progress of agriculture, the concept of fertilization has gradually changed to precision fertilization and reduction, a combination of organic and inorganic fertilizers, balanced nutrient fertilization, fertilization according to soil and plant needs, and improved fertilization methods such as bag-controlled slow-release fertilization, which are used to improve the utilization rate of fertilizer and increase soil fertility. It is of great significance for us to understand the characteristics of soil nutrient changes in peach orchards in the main peach-producing areas of Shandong Province over one decade and further discuss the quality of soil fertility to guide the management of soil nutrients in orchards (Figure 7).

### 4.2. Sample Collection and Analysis

In 2011 and mid-October 2021, representative peach orchards under local conventional management conditions were selected for soil collection. The sampling locations were Feicheng District, TA city and MY District, Linyi city. The ‘S’-shaped method was adopted in each orchard to randomly select 5–6 peach trees with basically the same growth, and a five-point sampling method was used to collect mixed soil samples of the 0–20 cm soil layer and 20–40 cm soil layer in the crown projection area. In total, 98 soil samples were collected in a traditional peach orchard, and 6 soil samples were collected in a demonstration peach orchard, i.e., a total of 104 soil samples. The collected soil was placed under natural conditions and air-dried, ground, sieved through a 100-mesh sieve, marked, mixed and bagged for backup.

Under the condition of a soil water ratio of 1:2.50 (*w*/*v*), soil pH was analysed with a pH electrode (PHS-3C, Shanghai, China). The SOM content was determined by the potassium dechromate-external heating method; soil AN in soil was determined by the diffusion method; soil AP (0.5 mol·L^−1^ Na_2_CO_3_ extraction) was determined by molybdenum antimony resistance colorimetry; and soil AK (1 mol·L^−1^ neutral NH_4_OAc extraction) was determined by the flame photometer method.

### 4.3. Data Analysis

SPSS software was used to perform the statistical analysis of the data [58], LSD was used for single-factor analysis of variance and difference significance analysis, and Origin Pro2021 was used to perform index correlation analysis. The *p* < 0.05 level was used for significance testing in all statistical analyses. Microsoft Excel 2019 and GraphPad Prism 8.0.1 software were used for figure drawing.

### 4.4. Nutrient Evaluation and Parameter Calculation Method

China’s second soil census standard was used in this study [59]. The nutrient content of organic fertilizers and chemical fertilizers was calculated based on the nutrient content reported on the fertilizer packaging bags used by farmers. The fertilization amount of the demonstration garden was two packs of controlled and slow-release fertilizer per tree (95 g per pack, urea, diammonium phosphate and potassium sulfate were mixed in a mass ratio of 41:14:40).

Based on the fuzzy comprehensive evaluation method and the principle of membership function and correlation analysis, the fertility quality of the peach orchards in the two production areas was comprehensively evaluated over one decade. According to the membership degree of each fertility index, four fertility factors, i.e., SOM, AN, AP and AK, were selected as evaluation indexes, and the ‘S’-type membership function was adopted:$$f(x)=\left\{\begin{array}{c} 1.0\\ 0.9(X-X1)/(X2-X1)+0.1\\ 0.1 \end{array}\right.$$

In the above formula, X1 and X2 correspond to the values of the membership function of each fertility index at 0.1 and 0.9, respectively. Since each individual fertility index has different effects on different crops and soils, the value at the inflection point depends on the actual situation. According to the membership function, the membership value Ni of each fertility index was calculated. This value is between 0.1 and 1. When the membership value is the maximum value of 1, the soil fertility quality is in the best condition; the minimum value of 0.1 indicates that the soil fertility level is low.

A correlation coefficient matrix of all fertility indexes was established, and then the average value of the correlation coefficient between each fertility indicator and other indicators was calculated. This average value accounted for the percentage of the sum of the average correlation coefficients of all fertility indicators and was the weight of the indicator (W_i_). Through the use of the fuzzy mathematical comprehensive evaluation method, an integrated fertility index (IFI) model was established for 4 soil fertility indicators of 52 orchards in the survey area in 2011 and 2021, and a comprehensive evaluation was carried out. The calculation formula is as follows:IFI = ∑W_i_ × N_i_

In the formula, W_i_ and N_i_ represent the respective weight and membership value of the ith index. The IFI was classified as follows: lower, IFI ≤ 0.2; low, 0.2 < IFI ≤ 0.4; medium, 0.4 < IFI ≤ 0.6; high, 0.6 < IFI ≤ 0.8; and higher, IFI > 0.8.

## 5. Conclusions

How to maximize the positive effects of fertilizers in agricultural production while minimizing their negative effects is a complex problem that is difficult to avoid in modern agriculture. The fundamental way to solve this problem is to establish a set of soil nutrient evaluation indicators and a scientific fertilizer application system in agricultural production. In this study, the temporal and spatial changes in the soil properties and their influential factors were studied in depth in 52 peach orchards, and the changes in soil fertility were effectively evaluated. The results showed that: (i) The input of nitrogen, phosphorus and potassium in organic fertilizer in 2021 was significantly higher than that in 2011, while the input of fertilizer in 2011 was significantly higher than that in 2021. Compared with traditional peach orchards, the demonstration peach orchards exhibited a significant downwards trend in both organic and fertilizer inputs. (ii) The SOM content and AP and AK contents in different soil layers increased significantly in 2021. The soil AN content was significantly lower in 2021 than in 2011. In the case of differences in the amount of fertilization, there was no significant difference in the soil nutrient content of the traditional peach orchards and demonstration peach orchards. (iii) The correlation analysis revealed a negative correlation between SOM and AN and a positive correlation with AP and AK. (iv) The calculation results of the integrated fertility index (IFI) values showed that the soil fertility quality was basically at a moderate level in 2011 and at a moderate to high level in 2021. Therefore, we recommend scientific fertilization of soil with different fertility levels. Through the demand for fertilizers by the tree body, the application of bag-controlled slow-release fertilizers is the main one; the proportion of chemical fertilizers is strictly controlled to meet the nutrition of the tree body, effectively improve soil fertility, and avoid the damage to the soil caused by large-scale chemical fertilizers, which is of great significance to soil nutrient management.

## Figures and Tables

**Figure 1 plants-12-01725-f001:**
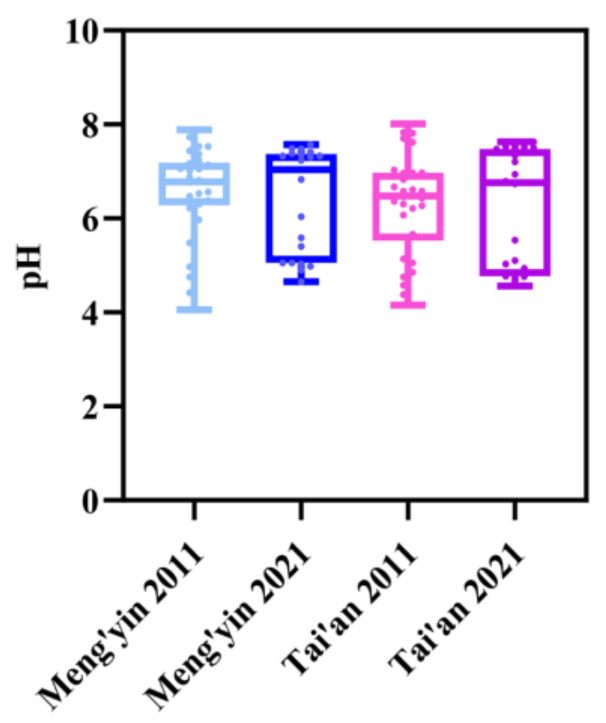
Changes in soil pH value in the main producing area of peach in Shandong in 2011 and 2021.

**Figure 2 plants-12-01725-f002:**
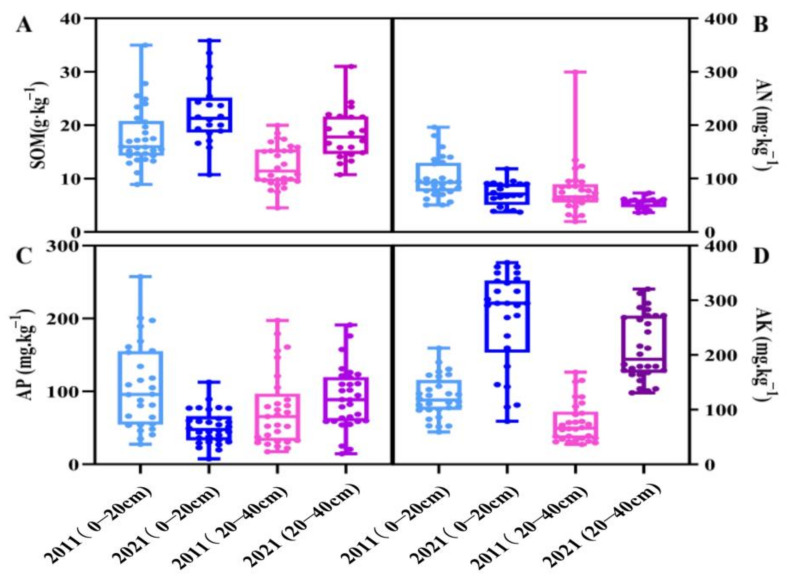
Changes in nutrient content of different soil layers in Tai’an in 2011 and 2021. (**A**): Soil organic matter (SOM); (**B**): alkaloids nitrogen (AN); (**C**): available phosphorus (AP); (**D**): available potassium (AK).

**Figure 3 plants-12-01725-f003:**
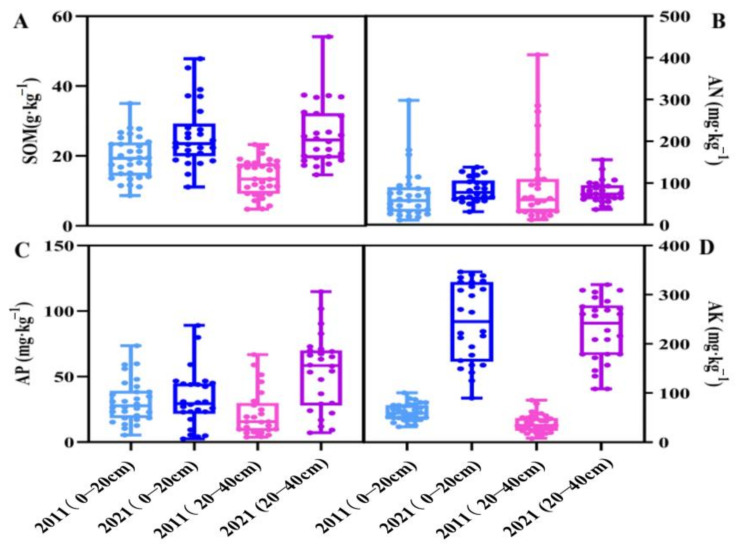
Changes in nutrient content of different soil layers in Meng’yin in 2011 and 2021. (**A**): SOM; (**B**): AN; (**C**): AP; (**D**): AK.

**Figure 4 plants-12-01725-f004:**
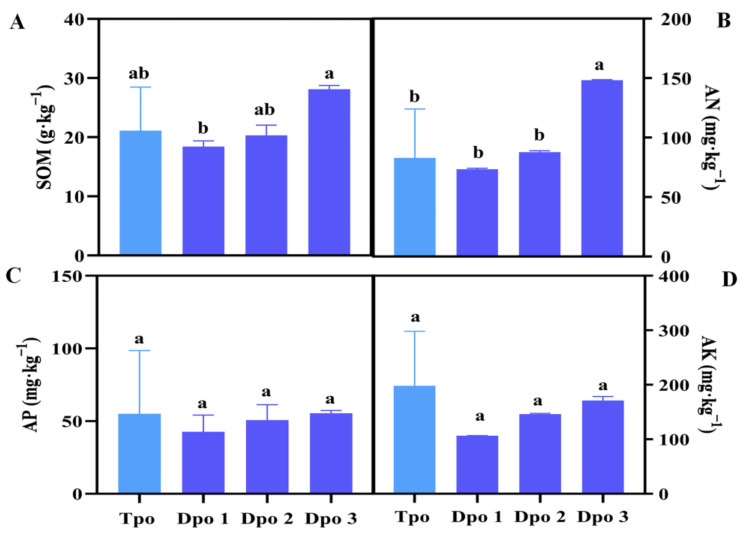
Comparison of the physical and chemical properties of the soil in the traditional park and the demonstration park. (**A**): SOM; (**B**): AN; (**C**): AP; (**D**): AK. Traditional peach orchard (Tpo); demonstration peach orchard (Dpo). Different letters indicate significant differences between different sites (*p* < 0.05), a and b are significant differences, ab is not significant differences.

**Figure 5 plants-12-01725-f005:**
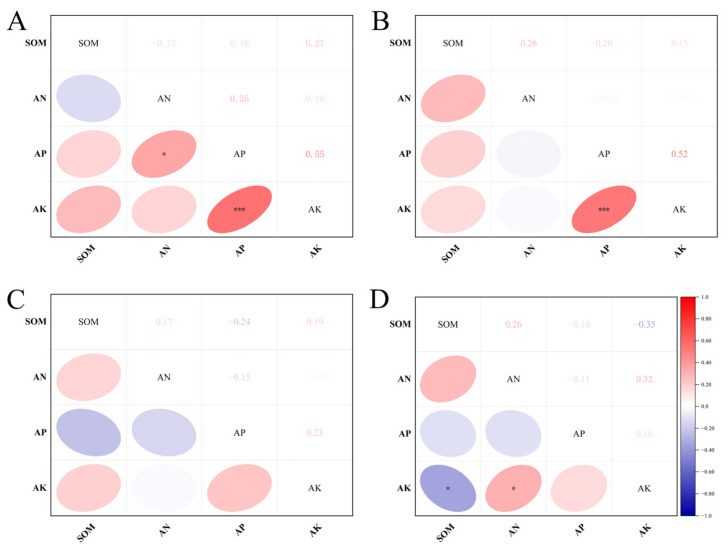
Pearson correlation of soil properties in 2011 and 2021. Color indicates the relevant direction (Blue = negative, red = positive). The *p*–value of dark colors is significant (* *p* < 0.05, *** *p* < 0.001). The correlation coefficients are shown in the panel. (**A**): 2011, 0–20 cm soil layer; (**B**): 2011, 20–40 cm soil layer; (**C**): 2021, 0–20 cm soil layer; (**D**): 2021, 20–40 cm soil layer. SOM, AN, AP, AK.

**Figure 6 plants-12-01725-f006:**
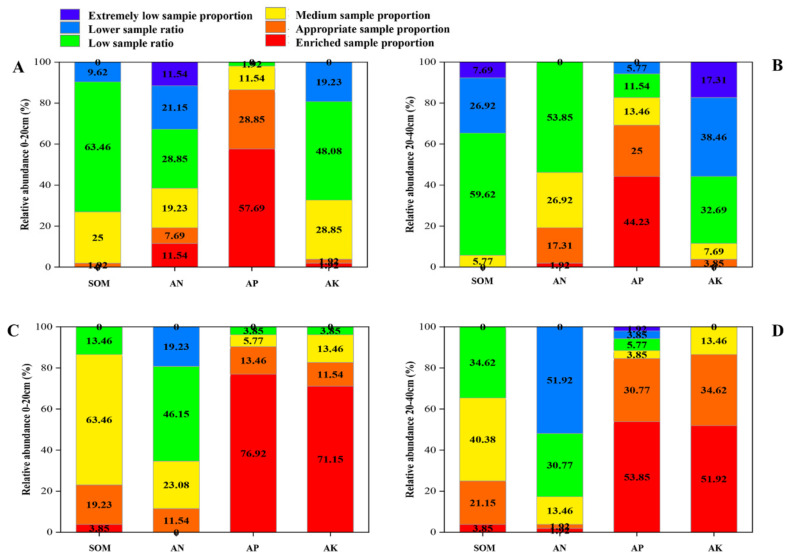
Classification of soil effective nutrient content in the main producing area of peach in Shandong in 2011 and 2021. (**A**): 2011, 0–20 cm soil layer; (**B**): 2011, 20–40 cm soil layer; (**C**): 2021, 0–20 cm soil layer; (**D**): 2021, 20–40 cm soil layer.

**Figure 7 plants-12-01725-f007:**
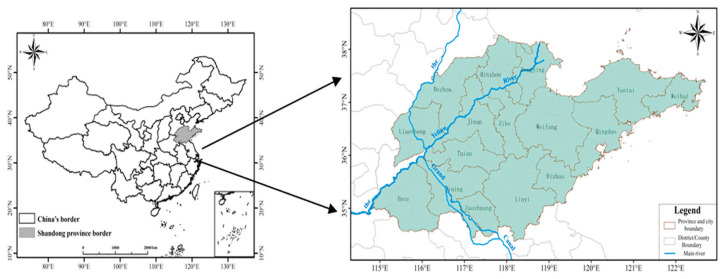
Geographical location map of Shandong Province, China.

**Table 1 plants-12-01725-t001:** Changes in the effective nutrient content of soil in different soil layers in in the main producing area of peach in Shandong in 2021.

Time	Parameters	Producing Area	Item	SOM (g·kg^−1^)	AN (mg·kg^−1^)	AP (mg·kg^−1^)	AK (mg·kg^−1^)
2011	0–20 cm	Tai’an	Average	17.82 ± 5.60 a	100.35 ± 37.70 a	107.13 ± 59.44 a	122.96 ± 37.81 a
			Range	8.9–35	50.5–196	27.53–257.41	59.12–212.36
		Meng’yin	Average	19.51 ± 5.8 a	71.68 ± 60.37 b	31.21 ± 16.26 b	60.87 ± 17.51 b
			Range	8.7–35	11.38–298	5.4–73.57	31.64–100.48
		Average		18.69 ± 5.79	85.76 ± 52.12	67.86 ± 57.20	90.90 ± 42.59
	20–40 cm	Tai’an	Average	12.43 ± 3.89 a	78.50 ± 51.56 a	76.53 ± 50.71 a	59.71 ± 37.05 a
			Range	4.5–19.97	19.6–299.6	17.27–197.36	36.56–168.64
		Meng’yin	Average	13.73 ± 5.13 a	94.16 ± 94.98 a	21.61 ± 17.75 b	37.83 ± 18.06 b
			Range	4.8–23.26	12.43–407.4	3.78–66.81	7.8–85.64
		Average		13.10 ± 4.58	86.47 ± 76.52	50.27 ± 47.14	56.63 ± 34.66
2021	0–20 cm	Tai’an	Average	25.61 ± 4.35 a	82.75 ± 28.90 a	90.54 ± 42.59 a	254.52 ± 90.08 a
			Range	19.32–35.83	37.33–148.87	14.61–191.24	93.64–372.41
		Meng’yin	Average	22.2 ± 5.15 b	83.73 ± 29.07 a	50.68 ± 29.23 b	259.29 ± 77.36 a
			Range	11.12–29	31.62–138.02	7.22–114.86	89.93–346.39
		Average		24.17 ± 4.96	83.16 ± 28.69	73.68 ± 42.18	256.54 ± 84.16
	20–40 cm	Tai’an	Average	23.8 ± 8.91 a	55.20 ± 16.09 b	51.16 ± 23.12 a	213.45 ± 58.36 a
			Range	10.72–38.29	31.15–100.68	7.60–112.59	132.05–315.42
		Meng’yin	Average	22.82 ± 5.04 a	79.68 ± 27.83 a	32.14 ± 23.18 b	236.06 ± 64.21 a
			Range	14.62–32.5	36.98–155.63	2.60–89.09	109.75–317.90
		Average		23.38 ± 7.47	65.56 ± 24.80	43.11 ± 24.80	233.02 ± 61.33

Notes: Values are means ± standard deviation and different letters within the same column denote significant differences (*p* < 0.05) among different sites. Soil organic matter (SOM), soil alkaloid nitrogen (AN), available phosphorus (AP) and available potassium (AK).

**Table 2 plants-12-01725-t002:** Differences in fertilizer inputs in the main producing area of peaches in Shandong in 2011 and 2021.

Time	Producing Area	Samples	Organic Fertilizer Input (kg·hm^−2^)			Chemical Fertilizer Input (kg·hm^−2^)			Total Input (kg·hm^−2^)		
			N	P_2_O_5_	K_2_O	N	P_2_O_5_	K_2_O	N	P_2_O_5_	K_2_O
2011	Tai’an	20	358.16 ± 15.74 b	544.68 ± 15.7 b	574.92 ± 11.04 b	682.61 ± 19.08 a	406.72 ± 18.25 a	306.65 ± 16.91 a	1040.77 ± 17.58 b	951.4 ± 16.78 b	881.57 ± 15.94 c
	Meng’yin	32	330.75 ± 10.60 b	555.06 ± 18.78 b	613.31 ± 11.18 b	528.14 ± 19.6 a	378.46 ± 18.71 a	350.89 ± 17.3 a	858.89 ± 12.66 b	933.52 ± 19.09 b	964.2 ± 17.72 b
	Total	52	344.45 ± 25.01 b	549.87 ± 27.97 b	594.11 ± 28.03 b	555.37 ± 29.37 a	397.59 ± 29.78 a	328.77 ± 28.24 a	899.82 ± 19.7 b	947.46 ± 29.27 b	922.88 ± 28.73 c
2021	Tai’an	20	627.82 ± 7.71 a	712.27 ± 19.51 a	764.37 ± 16.79 a	435.96 ± 19.56 b	323.24 ± 16.63 b	274.70 ± 15.49 b	1063.78 ± 22.74 a	1035.51 ± 10.38 a	1039.07 ± 18.87 a
	Meng’yin	26	599.34 ± 15.54 a	676.38 ± 18.23 a	699.77 ± 14.12 a	398.43 ± 25.56 b	267.45 ± 18.96 b	287.15 ± 15.61 b	997.77 ± 28.91 a	943.83 ± 18.26 b	986.92 ± 17.32 b
	Total	46	613.58 ± 17.33 a	694.32 ± 16.65 a	732.07 ± 16.23 a	417.19 ± 29.61 b	295.34 ± 18.68 b	280.92 ± 17.3 b	1030.77 ± 20.06 a	989.66 ± 33.29 b	1012.99 ± 24.47 ab
	Dpo 1	2	210 ± 6.21 c	135 ± 3.43 c	120 ± 8.35 c	41.24 ± 2.37 c	12.9 ± 1.21 c	40.2 ± 3.36 c	251.24 ± 8.43 c	147.9 ± 5.98 c	160.2 ± 6.37 d
	Dpo 2	2	217 ± 5.15 c	138 ± 5.08 c	123 ± 5.13 c	33.85 ± 2.23 c	10.62 ± 1.32 c	33 ± 2.07 c	250.85 ± 7.06 c	148.62 ± 3.65 c	156 ± 9.34 d
	Dpo 3	2	220 ± 5.87 c	141 ± 7.89 c	127 ± 9.43 c	36.93 ± 2.87 c	11.59 ± 1.53 c	36 ± 3.41 c	256.93 ± 11.48 c	152.59 ± 9.14 c	163 ± 9.78 d

Notes: Values are means ± standard deviation and different letters within the same column denote significant differences (*p* < 0.05) among different sites. Dpo (demonstration peach orchard).

**Table 3 plants-12-01725-t003:** The affiliation of various fertility indicators of different soil layers in 2011 and 2021.

Time	Parameters	Item	SOM (g·kg^−1^)	AN (mg·kg^−1^)	AP (mg·kg^−1^)	AK (mg·kg^−1^)
2011	0–20 cm	Average	0.36	0.45	0.82	0.25
		Range	0.1–0.85	0.1–1.0	0.11–1.0	0.1–1
		SD	0.17	0.34	0.24	0.23
		CV	0.472	0.756	0.292	0.92
	20–40 cm	Average	0.21	0.38	0.7	0.13
		Range	0.1–0.5	0.1–1.0	0.1–1.0	0.1–0.72
		SD	0.11	0.35	0.33	0.11
		CV	0.52	0.92	0.47	0.84
2021	0–20 cm	Average	0.52	0.48	0.79	0.85
		Range	0.12–1.0	0.1–1.0	0.1–1.0	0.1–1.0
		SD	0.22	0.33	0.29	0.27
		CV	0.42	0.68	0.36	0.32
	20–40 cm	Average	0.51	0.31	0.89	0.87
		Range	0.12–1.0	0.1–1.0	0.16–1.0	0.19–1.0
		SD	0.24	0.28	0.23	0.21
		CV	0.47	0.9	0.26	0.24

Notes: Standard deviation (SD), coefficient of variation (CV). Soil organic matter (SOM), soil alkaloid nitrogen (AN), available phosphorus (AP) and available potassium (AK).

**Table 4 plants-12-01725-t004:** Distribution ratio of orchard fertility level (%).

Time	Parameters	I	II	III	IV	V
2011	0–20 cm	0.00	7.69	38.46	40.38	13.46
	20–40 cm	0.00	3.85	9.62	67.3	19.23
2021	0–20 cm	3.85	34.62	51.92	9.62	0.00
	20–40 cm	1.92	21.15	65.38	9.62	1.92

## Data Availability

The data presented in this study are available on request from the corresponding author.

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
