# Peer review of "Dynamic Monitoring of Nutrition Inputs and Fertility Evaluation during a Decade in the Main Peach-Producing Areas of Shandong Province, China"

_plants, 2023, doi:10.3390/plants12081725_

Round 1
Reviewer 1 Report
The manuscript is well organized, detail oriented and easy to read and understand. The methods are well described so can be replicated. The results are well presented and the conclusion addresses the key findings.
Author Response
Thank you for your letter and the Reviewers’ comments concerning our manuscript
entitled “Dynamic monitoring of nutrition inputs and fertility evaluation during a decade in the main peach-producing areas of Shandong Province, China”. (Manuscript ID: 2257138). Those comments are all valuable and very helpful for revising and improving our paper and have great guiding significance to our researches. We have studied the comments carefully and have made corresponding corrections which can meet with your approval hopefully. The main corrections in the paper and the responses to the Reviewer’s comments are as following:
Thank you for taking time to review my article and thank you for your recognition of my article.

Reviewer 2 Report
The paper has potential but must be properly revised. Please see my suggestions regarding this manuscript:
Please read and apply the Instructions for authors. Begin with the number of the words allowed for the Abstract (which is too long).
In the entire manuscript please revise Abbreviations as the Instructions for authors must be checked and respected “Acronyms/Abbreviations/Initialisms should be defined the first time they appear in each of three sections: the abstract; the main text; under the first figure or table. When defined for the first time, the acronym/abbreviation/initialism should be added in parentheses after the written-out form”. Revise the entire manuscript in this regard.
L 38. “Soil nutrients and enzymology…” I suggest checking and referring to Samuel A.D., Brejea R., Domuta C., Bungau S., Cenusa N., Tit D.M. Enzymatic indicators of soil quality. J. Environ. Prot. Ecol. 2017, 18(3), 871-878 and https://doi.org/10.37358/RC.17.10.5864
L43. Why 5 references [6-10] for a single sentence? Please provide recent references and do not exceed them.
L56. “climate” influence on the orchards must be detailed and referenced. I suggest referring to https://doi.org/10.1007/s11356-019-04214-1
L99. Please improve the aim of your research by adding the novelty/special aspects it brings to the field as to increase the potential readers’ interest.
Comprehensive, good Results part.
However, to a better aspect, at Tables 1 and 2 (and 3 if you consider, even this table is well fitted on the page). I suggest after the line of the head of the table add one more row (merged for all the columns) – where to insert 2011. When finishing the info about 2011, insert one more row, where inserting 2021. In this way you can remove the first column (Time) and you can reposition your table in a better way as to fit the space. Moreover, under the Table please explain all abbreviations used in the head of the table (as the Instructions for authors also mention).
Discussion. What about using nanotechnology in agriculture. Also develop this idea by checking https://doi.org/10.1016/j.chemosphere.2021.132533 nanotechnology and nano chemistry allowing a different approach for fertilizers in agriculture.
L347. Which investigation? Have been they questioned? If yes, there is much more that must be written.
After L353. As the last paragraph of the Discussion section, you should highlight the strengths and the weakness of your results.
L357. A satellite image of the location would be relevant as figure.
Subsection 4.3. Statistical software must be also referenced. I suggest checking https://libguides.library.kent.edu/statconsulting/software-help and proceed consequently.
L400-401. Reference must be added in the brackets like all other references, not as link in the main text.
References section. Please check the Instructions for authors regrading References requested info and insert them in the MDPI style.
Author Response
Response to Reviewer 2 Comments
Thank you for your letter and the Reviewers’ comments concerning our manuscript
entitled “Dynamic monitoring of nutrition inputs and fertility evaluation during a decade in the main peach-producing areas of Shandong Province, China”. (Manuscript ID: 2257138). Those comments are all valuable and very helpful for revising and improving our paper and have great guiding significance to our researches. We have studied the comments carefully and have made corresponding corrections which can meet with your approval hopefully. The main corrections in the paper and the responses to the Reviewer’s comments are as following:
Responses to the Reviewer’s comments:
Point 1: Please read and apply the Instructions for authors. Begin with the number of the words allowed for the Abstract (which is too long).
Response 1: Thanks for your valuable question. Considering the Reviewer’s suggestion, we have condensed the abstract. (lines10-32)
Point 2: In the entire manuscript please revise Abbreviations as the Instructions for authors must be checked and respected “Acronyms/Abbreviations/Initialisms should be defined the first time they appear in each of three sections: the abstract; the main text; under the first figure or table. When defined for the first time, the acronym/abbreviation/initialism should be added in parentheses after the written-out form”. Revise the entire manuscript in this regard.
Response 2: Thanks for your valuable question. Considering the Reviewer’s suggestion, we have changed the acronym. (lines26-28, 40,41, 110, 111, 144-147, 149-151, 165-167, 174-176, 184-187, 203-205, 223, 224, 263, 295-297, 424, 472)
Point 3: L 38. “Soil nutrients and enzymology…” I suggest checking and referring to SamuelA.D., BrejeaR., DomutaC., BungauS., CenusaN., TitD.M. Enzymatic indicators of soil quality. J. Environ. Prot. Ecol. 2017, 18(3), 871-878 and https://doi.org/10.37358/RC.17.10.5864
Response 3: Thanks for your valuable question. Considering the Reviewer’s suggestion, we have
already referenced that document and cited it in the article. (line39)
Point 4: L43. Why 5 references [6-10] for a single sentence? Please provide recent references and do not exceed them.
Response 4: Thanks for your valuable question. Considering the Reviewer’s suggestion, we have We have deleted some references and kept the most recent references. (line44)
Point 5: L56. “climate” influence on the orchards must be detailed and referenced. I suggest referring to https://doi.org/10.1007/s11356-019-04214-1
Response 5: Thanks for your valuable question. We refer to the literature provided by the reviewers to supplement the”climate" aspect, For example, climate change will have an impact on the soil structure and soil fertility of orchards, which will further affect the annual reproductive cycle of fruit trees: the nutrient rest period of fruit trees is shortened, resulting in early flowering of fruit trees. (line57-60)
Point 6: L99. Please improve the aim of your research by adding the novelty/special aspects it brings to the field as to increase the potential readers’ interest.
Response 6: Thanks for your valuable question. Considering the Reviewer’s suggestion, we have supplemented the content of 99 lines, highlighting the novelty of this research field. It can more intuitively reflect the characteristics of soil fertility changes in Taoyuan 10 years ago and ten years later, and compare with the demonstration Taoyuan to highlight the effects of weight loss and fertilizer control technology, and provide a strong basis for Taoyuan's next fertilization management. (line98-101)
Point 7: However, to a better aspect, at Tables 1 and 2 (and 3 if you consider, even this table is well fitted on the page). I suggest after the line of the head of the table add one more row (merged for all the columns) – where to insert 2011. When finishing the info about 2011, insert one more row, where inserting 2021. In this way you can remove the first column (Time) and you can reposition your table in a better way as to fit the space. Moreover, under the Table please explain all abbreviations used in the head of the table (as the Instructions for authors also mention).
Response 7: Thanks for your valuable question. Considering the Reviewer’s suggestion, we have re-adjusted the table to make it easier for readers to read. At the same time, the table below explains all abbreviations used in the header.
Point 8: Discussion. What about using nanotechnology in agriculture. Also develop this idea by checking https://doi.org/10.1016/j.chemosphere.2021.132533 nanotechnology and nano chemistry allowing a different approach for fertilizers in agriculture.
Response 8: Thanks for your valuable question. Considering the Reviewer’s suggestion, we have added references to nano fertilizers to the discussion. (line328)
Point 9: L347. Which investigation? Have been they questioned? If yes, there is much more that must be written.
Response 9: Thanks for your valuable question. Considering the Reviewer’s suggestion, We revised the meaning of that sentence. According to our daily visits, we learned that farmers often pay more attention to the application of a large number of elements and often easily ignore the application of trace elements. (line388,389)
Point 10: After L353. As the last paragraph of the Discussion section, you should highlight the strengths and the weakness of your results.
Response 10: Thanks for your valuable question. Considering the Reviewer’s suggestion, we have revised and supplemented the last paragraph of the article. This research focuses on 52 Peach orchard, the main peach producing area in Shandong. The spatiotemporal changes of soil traits and their influencing factors were studied in depth, and the changes in soil fertility were effectively evaluated. It was found that under the management conditions of fertilizer control and weight loss, the soil nutrient content of the demonstration park was not significantly different from that of the traditional Peach orchard. It is concluded that water-saving and efficiency-enhancing measures have significantly improved soil nutrients. It provides theoretical support for future research on Peach orchard management technology. However, there have been some changes in whether reducing fertilization affects the content of microorganisms in the soil. Whether it promotes some microbial flora, such as those involved in soil nitrogen fixation and phosphorus and potassium decomposition, to become active in the soil, thereby improving soil fertility, we still have not reached a conclusion. These are still the goals of our team's next research. (lines 372-384)
Point 11: L357. A satellite image of the location would be relevant as figure.
Response 11: Thanks for your valuable question. Considering the Reviewer’s suggestion, we have added satellite imagery to the method. (line419)
Point12: Subsection 4.3. Statistical software must be also referenced. I suggest checking https://libguides.library.kent.edu/statconsulting/software-help and proceed consequently.
Response 12: Thanks for your valuable question. Considering the Reviewer’s suggestion, in this paper, we refer to the statistical software provided by the reviewer. (line439)
Point 13: L400-401. Reference must be added in the brackets like all other references, not as link in the main text.
Response 13: Thanks for your valuable question. Considering the Reviewer’s suggestion, we have changed the links in the article to a reference format. (line447)
Point14: References section. Please check the Instructions for authors regrading References requested info and insert them in the MDPI style.
Response 14: We have checked and modified all the references in accordance with the document format requirements of MDPI.

Reviewer 3 Report
The article is related to monitoring of nutrient inputs and fertilizer evaluation during a decade in the main peach-producing areas in China. One of the goals of this article was to study the changes in soil properties over one decade. The authors determined the pH, soil organic matter, alkaloid nitrogen, available phosphorus and available potassium in soil samples in 2011 and 2021. Why was the change over time not studied? For example, every year or every 2 years?
The research methodology is not clear to me. How often was fertilizer added during this 10 years? What was the organic fertilizer? Were the peaches fertilized with organic and inorganic fertilizers?
The second goals of this article was to explore the causes of the soil property changes/ however, I did not found a strong explanation of the changes in soil during 10 years.
In subsection 2.3 authors presented a correlation between different parameters, but no explanation of what it means.
In conclusions the authors recommend the scientific use of fertilizers, however did not explain what it means.
In my opinion, the article has no scientific value or I failed to notice it. I do not understand for what authors conduct this study and how the results can be used.
Author Response
Response to Reviewer 3 Comments
Thank you for your letter and the Reviewers’ comments concerning our manuscript
entitled “Dynamic monitoring of nutrition inputs and fertility evaluation during a decade in the main peach-producing areas of Shandong Province, China”. (Manuscript ID: 2257138). Those comments are all valuable and very helpful for revising and improving our paper and have great guiding significance to our researches. We have studied the comments carefully and have made corresponding corrections which can meet with your approval hopefully. The main corrections in the paper and the responses to the Reviewer’s comments are as following:
Responses to the Reviewer’s comments:
Point 1: The article is related to monitoring of nutrient inputs and fertilizer evaluation during a decade in the main peach-producing areas in China. One of the goals of this article was to study the changes in soil properties over one decade. The authors determined the pH, soil organic matter, alkaloid nitrogen, available phosphorus and available potassium in soil samples in 2011 and 2021. Why was the change over time not studied? For example, every year or every 2 years?
Response 1: Thanks for your valuable question. What we studied is the changes in nutrient input and soil fertility in the main peach producing areas of Shandong Province, China over the past ten years. Due to the large number of resources involved and the huge workload, we did not investigate every year or every two years.
Point 2: The research methodology is not clear to me. How often was fertilizer added during this 10 years? What was the organic fertilizer? Were the peaches fertilized with organic and inorganic fertilizers?
Response 2: Thank you for your valuable question. Among the 52 peach orchards, it mainly includes peach orchards under conventional management and peach orchards under scientific management and fertilization management. Conventional management is mainly based on farmers' field fertilization habits, and farmers often pay attention to the large-scale application of chemical fertilizers. The scientific management of peach orchards refers to the fertilization method of bag-controlled slow-release fertilizer, strict control of the fertilization ratio of chemical fertilizers, and mainly attaches importance to the application of organic fertilizers. Organic fertilizer is mainly based on earthworm manure and cattle and sheep manure.
Point 3: The second goals of this article was to explore the causes of the soil property changes/ however, I did not found a strong explanation of the changes in soil during 10 years.
Response 3: Thanks for your valuable question. Because we have changed the way of fertilization in the past ten years, by reducing fertilizer and controlling fertilizer to effectively improve soil fertility, we found that soil organic matter, available phosphorus, available potassium, and soil fertility have been significantly improved compared with ten years ago, and there is no significant difference in nutrient content between traditional peach orchard and demonstration peach orchard under the management of weight loss and fertilizer control. This further shows that our weight loss and fertilizer control measures play an important role in reducing fertilizer application and improving soil fertility.
Point 4: In conclusions the authors recommend the scientific use of fertilizers, however did not explain what it means.
Response 4: Thanks for your valuable question. Considering the reviewer's opinion, we explain the scientific application of fertilizer in the conclusion. “Therefore, we recommend scientific fertilization of soil with different fertility levels. Through the demand for fertilizers by the tree body, the application of bag-controlled slow-release fertilizers is the main one, and the proportion of chemical fertilizers is strictly controlled to meet the nutrition of the tree body, effectively improve soil fertility, and avoid the damage caused by large-scale chemical fertilizers to the soil, which is of great significance to soil nutrient management”. (line499-504)
Point 5: In my opinion, the article has no scientific value or I failed to notice it. I do not understand for what authors conduct this study and how the results can be used.
Response 5: Thanks for your valuable question. We can see that the large-scale application of fertilizers in recent years has caused adverse effects on soil fertility, and the decrease in soil fertility will further affect the growth of fruit trees. This paper mainly analyzes the evolution of soil properties under conventional fertilization and scientific fertilization management. The results show that under the scientific reduction of fertilizer application, soil fertility has been well improved.It is of great significance to promote high yield and high quality through the application of a small amount of fertilizer in the production and management of fruit trees.

Reviewer 4 Report
This is an intriguing study because it sheds light on the evolution of soil properties in peach orchards under various fertilization regimes and aids in the design and adjustment of management fertilization methods. The manuscript as a whole is well written and presented to the reader. The introduction is based on recent findings, and "Materials and Methods" are thoroughly described. Statistical analysis has been used to validate the results. The reader will find most of the tables and figures to be clear and understandable. The main conclusions and ideas are adequate and justified in general. A few minor changes are suggested (please see also the attached file) in the following lines:
Line 149, Table 1.: Please use horizontal lines to distinguish each soil horizon. The reader can then compare layers and regions more easily.
Line 160: In figure 4b AN in the traditional park differed significantly when the amount of fertilizer differed.
Line 207, Table 2: Please increase the width of the columns. Table format confuses the reader.
Line 273, Table 3: Please use horizontal lines to distinguish each soil horizon. The reader can then compare layers and years more easily.
Lines 287-288: Is that the main reason why the pH value remained constant? What about soil's buffer capacity?
Line 291: Bottom soil profiles in ponds are commonly referred to as "bottom soil." You could also use phrases like "deeper soil layers." In any case, there are two soil layers in this study. As a result, the 0-20 and 20-40 cm layers can also be used.
Lines 293-294: “At the same time, the topsoil is the best place for fertilization, resulting in this layer being the most nutrient rich”. Which is the best place for fertilization after topsoil? The meaning is unclear. Please rewrite the sentence.
Lines 327-328: 56]. …….In areas with high soil fertility, crop yields and nutrient concentrations are significantly higher than those in areas with low soil fertility [57]……..
This sentence seems not to be related to the discussion.
Lines 333-337: Please add references. Please use other related works to support your suggestions.
Based on the above, the manuscript could be considered for publication.

Author Response
Response to Reviewer 4 Comments
Thank you for your letter and the Reviewers’ comments concerning our manuscript
entitled “Dynamic monitoring of nutrition inputs and fertility evaluation during a decade in the main peach-producing areas of Shandong Province, China”. (Manuscript ID: 2257138). Those comments are all valuable and very helpful for revising and improving our paper and have great guiding significance to our researches. We have studied the comments carefully and have made corresponding corrections which can meet with your approval hopefully. The main corrections in the paper and the responses to the Reviewer’s comments are as following:
Responses to the Reviewer’s comments:
Point 1: Line 149, Table 1.: Please use horizontal lines to distinguish each soil horizon. The reader can then compare layers and regions more easily.
Response 1: Thanks for your valuable question. Considering the Reviewer’s suggestion, We have added horizontal lines to Table 1 to better distinguish each soil layer and make it easier for readers to compare soil layers and areas. (line163)
Point 2: Line 160: In figure 4b AN in the traditional park differed significantly when the amount of fertilizer differed.
Response 2: Thanks for your valuable question. Considering the Reviewer’s suggestion, we have revised the expression of Result 2.2 in the article. (line177-179)
Point 3: Line 207, Table 2: Please increase the width of the columns. Table format confuses the reader.
Response 3: Thanks for your valuable question. Considering the Reviewer’s suggestion, We have modified the format in Table 2 to make it easier for readers to read. (line260)
Point 4: Line 273, Table 3: Please use horizontal lines to distinguish each soil horizon. The reader can then compare layers and years more easily.
Response 4: Thanks for your valuable question. Considering the Reviewer’s suggestion, we have added horizontal lines to Table 3 to better distinguish each soil layer and make it easier for readers to compare soil layers and years. (line293)
Point 5: Lines 287-288: Is that the main reason why the pH value remained constant? What about soil's buffer capacity?
Response 5: Thanks for your valuable question. Application of fertilizers is one of the causes of soil acidification (Hoyt & Hennig 1982). The acidification of soil by N fertilizer is caused by transformation of nitrogen in soil. The uptake of N as ammonium in the crop also contributes to soil acidification (Malhi et al. 1998). Our study found that the use of chemical fertilizers decreased, the content of organic fertilizers increased, and the nitrogen input significantly decreased during the decade, which has important significance for reducing soil acidification.
The buffer capacity of soil to acid or alkali is an important index of soil quality evaluation. In order to evaluate soil acidification process and predict acidification trend, it is necessary to study soil acid buffer capacity deeply. Our investigation found no significant change in pH, so we didn't go into it
Point 6: Line 291: Bottom soil profiles in ponds are commonly referred to as "bottom soil." You could also use phrases like "deeper soil layers." In any case, there are two soil layers in this study. As a result, the 0-20 and 20-40 cm layers can also be used.
Response 6: Thanks for your valuable question. Considering the Reviewer’s suggestion, we have changed the "topsoil" in line 301 to 0-20cm and the "bottom soil" to 20-40cm. (line314)
Point 7:Lines 293-294: “At the same time, the topsoil is the best place for fertilization, resulting in this layer being the most nutrient rich”. Which is the best place for fertilization after topsoil? The meaning is unclear. Please rewrite the sentence.
Response 7: Thanks for your valuable question. Considering the Reviewer’s suggestion,We re-revised the meaning of that sentence. “The surface of the soil is 0-20cm, which is the interface between the soil and the atmosphere. The soil formation effect is the strongest and the time is the longest.It has strong bioaccumulation, contains more humus, and has high fertility. For cultivated soil, it is also the location of fertilization, which will naturally cause it to be the most nutrient-rich.” (line315-319)
Point 8: Lines 327-328: 56]. …….In areas with high soil fertility, crop yields and nutrient concentrations are significantly higher than those in areas with low soil fertility [57]……..
This sentence seems not to be related to the discussion.
Response 8: Thanks for your valuable question. Considering the Reviewer’s suggestion, we deleted the description of that lines 327-328: 56]. …….In areas with high soil fertility, crop yields and nutrient concentrations are significantly higher than those in areas with low soil fertility [57]……... (line356-359)
Point 9: Lines 333-337: Please add references. Please use other related works to support your suggestions.
Response 9: Thanks for your valuable question. Considering the Reviewer’s suggestion, we added references. (Lines 368)

Round 2
Reviewer 2 Report
The authors improved their paper.
Reviewer 3 Report
I recommend to publish this paper